# Clinically-Guided Counterfactuals (C$^3$): Physics and Pathology-Aware Augmentation and Evaluation for Robust Medical Imaging Models

## Abstract

Clinical deployment of imaging AI remains fragile: routine distribution shifts—scanner vendor and reconstruction kernel, MRI protocol updates, dose and slice profile changes, patient positioning and demographics, and device optics—can degrade performance in ways that standard leaderboards and generic augmentations fail to predict. We ask whether robustness and calibration can be improved, without compromising clinical validity, by training and evaluating models against *label-preserving, clinically grounded counterfactuals*. We introduce *Clinically-Guided Counterfactuals (C$^3$)*, a framework that (i) unifies physics-informed acquisition perturbations with tightly constrained, pathology-preserving semantic edits; (ii) screens all counterfactuals through a conservative validity gate; and (iii) reports *shift-stable utility*, a worst-case case-level score complementary to AUROC, Dice, ECE, and Brier. Across chest X-ray (CheXpert→MIMIC-CXR), MS brain MRI segmentation (multi-site→held-out site), and diabetic retinopathy grading (EyePACS→Messidor-2), C$^3$ delivers consistent OOD gains (e.g., macro-AUROC +0.035 on CXR; lesion-wise Dice +0.044 on MRI; DR AUROC +0.036), tighter calibration, reduced prediction volatility under realistic shifts, and interpretable robustness diagnostics suitable for deployment checks.

## 1 Introduction

Deep models for medical imaging often underperform when faced with routine distribution shifts (4), from reconstruction kernels and protocol changes to illumination and sensor variation (3). Standard data augmentation and aggregate metrics (e.g., AUROC, Dice) offer limited visibility into per-case worst-case behavior, which is crucial for safe deployment.

We propose to reframe robustness around each image's *clinically admissible neighborhood*: acquisition-realistic and pathology-preserving variants for which the diagnostic label or lesion mask should remain invariant. Concretely, for input $x$ with label $y$ (classification) or mask $m$ (segmentation), we define transformations $\{t_\phi\}_{\phi \in \Phi}$ and the counterfactual neighborhood $\mathcal{N}(x) = \{ t_\phi(x) : \phi \in \Phi,\ \text{label}(t_\phi(x)) = y \text{ or } \text{mask}(t_\phi(x)) = m \}$. Within this neighborhood, models should maintain consistent predictions and stable calibration.

**Main Contributions:**

- *Clinically-Guided Counterfactuals (C$^3$):* a framework combining (i) physics-aware acquisition perturbations, (ii) a pathology-preserving editor for small but clinically plausible semantic/context edits, and (iii) a *validity gate* that filters counterfactuals via conservative equivalence tests and sparse human anchors.

- *Training objective and evaluation:* a regularized objective that enforces prediction consistency across $\mathcal{N}(x)$, and a deployment-facing *shift-stable utility* metric summarizing

worst-case calibrated performance per exam, reported alongside AUROC, Dice, ECE, and Brier.

- *Evidence across three modalities/tasks:* CXR classification, MS MRI lesion segmentation, and DR grading show consistent OOD improvements, better calibration, and reduced volatility under scanner/protocol/camera shifts, with ablations demonstrating the complementary roles of physics transforms, the editor, and the validity gate.

## 2 Methodology

### 2.1 Counterfactual neighborhoods and consistency objective

For classification with model $f_\theta$, let $p_\theta(\cdot \mid x)$ denote predictive probabilities. For segmentation, let $\hat{m}_\theta(x)$ be the predicted mask. We train with

$$\mathcal{L} = \mathcal{L}_{\text{sup}}(f_\theta(x), y) \; + \; \lambda \, \mathbb{E}_{x' \sim \mathcal{N}(x)} \big[ D(f_\theta(x), f_\theta(x')) \big], \tag{1}$$

where $D = D_{\text{KL}}(p_\theta(\cdot \mid x) \,\|\, p_\theta(\cdot \mid x'))$ for classification and

$$D = 1 - \frac{2\langle \hat{m}_\theta(x), \hat{m}_\theta(x') \rangle + \epsilon}{\|\hat{m}_\theta(x)\|_1 + \|\hat{m}_\theta(x')\|_1 + \epsilon} \tag{2}$$

for segmentation.

For evaluation, we compute per-case *shift-stable utility*

$$U(x) = \min_{x' \in \mathcal{N}(x)} s\big(f_\theta(x'), y\big) \quad \text{or} \quad U(x) = \min_{x' \in \mathcal{N}(x)} \text{Dice}\big(\hat{m}_\theta(x'), m\big), \tag{3}$$

which lower-bounds performance under admissible clinical variation.

### 2.2 Physics-aware acquisition perturbations

We implement modality-specific operators that approximate routine acquisition changes while preserving labels/masks:

- *CT/CXR:* Poisson thinning before FBP for low dose; kernel sharpness variation (soft↔sharp); blur consistent with thicker slices/partial volume; mild beam hardening / scatter shifts of HU distributions.
- *MRI:* $B_0/B_1$ bias fields; sequence-appropriate Rician/non-central-$\chi$ noise; contrast modulation via Bloch-informed lookups (TE/TR/flip); slice-profile broadening.
- *Fundus:* Illumination geometry, vignetting, and sensor-pattern perturbations matched to camera response, preserving microaneurysms and exudates.

### 2.3 Pathology-preserving editor

A diffusion backbone is fine-tuned with weak supervision (reports/labels) to produce low-amplitude, clinically plausible edits (e.g., rib-shadow contrast, subtle effusion haze, projection geometry; illumination and vessel-contrast tweaks in fundus). Two soft constraints keep edits near the clinical manifold: (i) a lesion-mask consistency penalty discouraging changes to annotated pathology, and (ii) a text–image agreement term over a curated pathology vocabulary to stabilize the global clinical description.

### 2.4 Validity gate

Before inclusion in training/evaluation, counterfactuals must satisfy conservative tests:

- *Classification:* $\|p_\theta(\cdot \mid x) - p_\theta(\cdot \mid x')\|_\infty \leq \delta$ and matching predicted class argmax.
- *Segmentation:* teacher/consensus masks must meet $\text{IoU}(m^\star(x), m^\star(x')) \geq \tau$.

Thresholds $(\delta, \tau)$ are set using radiologist-audited anchors.

## 2.5 Experimental settings

We hold architectures/optimizers/schedules fixed to isolate $C^3$:

1. *CXR classification:* DenseNet-121 on CheXpert (1), OOD on MIMIC-CXR (2); 5 findings (Atelectasis, Cardiomegaly, Consolidation, Edema, Pleural Effusion).

2. *MS MRI segmentation:* 3D U-Net on a multi-site cohort; OOD evaluation on a held-out site; lesion- and volume-wise Dice.

3. *Fundus DR grading:* EfficientNet-B3 on EyePACS; OOD on Messidor-2; AUROC for referable DR and ECE.

Baselines: standard augmentations, RandAugment/AugMix variants, and TTA.

## 3 Results and Discussion

**Chest X-ray (CheXpert→MIMIC-CXR).** $C^3$ improves macro-AUROC from $0.864$ (STDAUG) and $0.872$ (AUGMIX) to $0.907$, with consistent per-pathology gains (e.g., Edema $0.903 \rightarrow 0.935$, Consolidation $0.842 \rightarrow 0.887$, Effusion $0.918 \rightarrow 0.944$). Calibration improves (ECE $5.7\% \rightarrow 2.9\%$; Brier $-13.4\%$ relative). Neighborhood agreement rises $0.73 \rightarrow 0.86$, and shift-stable utility increases $0.782 \rightarrow 0.846$, indicating stronger worst-case performance under clinically realistic perturbations. Ablations show physics transforms and the editor contribute additively, while removing the validity gate superficially boosts ID AUROC yet harms worst-case utility and calibration, exposing hidden label drift.

**MS MRI segmentation (multi-site→held-out).** Mean lesion-wise Dice increases from $0.598$ (STDAUG) and $0.624$ (AUGMIX) to $0.668$ with $C^3$; small-lesion recall ($< 10\,\mathrm{mm}^3$) improves $0.521 \rightarrow 0.603$ with $17.8\%$ fewer false negatives. Volume-wise Dice rises $0.706 \rightarrow 0.744$, and volume calibration tightens (slope $0.81 \rightarrow 0.93$). Under simulated slice-thickness increase ($1.0\,\mathrm{mm} \rightarrow 3.0\,\mathrm{mm}$), Hausdorff-95 decreases from $8.9\,\mathrm{mm}$ to $7.3\,\mathrm{mm}$, whereas non-$C^3$ baselines exceed $9.5\,\mathrm{mm}$. Physics-only (Dice $0.653$) and editor-only ($0.639$) trail the full model ($0.668$).

**Fundus DR (EyePACS→Messidor-2).** AUROC improves $0.842 \rightarrow 0.878$ over STDAUG (and $0.855 \rightarrow 0.878$ vs. TTA), while ECE halves ($4.2\% \rightarrow 2.0\%$). Prediction flips under admissible illumination/camera shifts drop from $14.7\%$ to $6.8\%$. The $10^{\mathrm{th}}$ percentile of per-patient shift-stable utility increases $0.763 \rightarrow 0.820$, and OOD AUROC CIs shrink by $24\%$, reflecting reduced variance across nuisance factors.

**Interpretability and deployment diagnostics.** $C^3$ attenuates failure modes aligned with radiologist intuition: e.g., CXR reliance on rib-shadow/illumination artifacts; MRI small-lesion under-segmentation exacerbated by bias fields or thicker slices; fundus sensitivity to vascular glare. Per-exam robustness profiles provide actionable diagnostics for model cards and site readiness checks.

**Takeaways.** (1) Physics-aware operators anchor robustness to acquisition realities; (2) small, clinically plausible edits broaden coverage of nuisance semantics without drifting labels; (3) a strict validity gate is essential to avoid training on mislabeled counterfactuals; (4) worst-case, case-level reporting (*shift-stable utility*) complements aggregate metrics and reveals deployment-relevant volatility.

## 4 Conclusion

$C^3$ reframes robustness around clinically grounded counterfactual neighborhoods, aligning both training and evaluation with how images vary across scanners, protocols, and devices. The approach consistently improves OOD accuracy, calibration, and worst-case stability across CXR, MRI, and fundus tasks, while surfacing interpretable diagnostics for deployment. Because $C^3$ composes with standard datasets and models, it can be adopted without architectural changes.

# 5  Future Directions

- *Prospective and site-onboarding studies:* Evaluate C$^3$ in prospective multi-site rollouts and during scanner/protocol onboarding to quantify reductions in drift-related incidents.
- *Expanded modalities and tasks:* Extend physics operators and editors to ultrasound, mammography, and pathology WSIs; explore detection and multi-label settings.
- *Human-in-the-loop validity:* Incorporate lightweight radiologist spot-audits and active sampling to calibrate $(\delta, \tau)$ and prioritize hard neighborhoods.
- *Fairness & subpopulation robustness:* Construct neighborhoods that target demographic/device subgroups, pairing shift-stable utility with stratified fairness metrics.
- *Uncertainty and calibration:* Combine C$^3$ with deep ensembles/temperature scaling under neighborhood perturbations; track case-level reliability diagrams.
- *Efficiency:* Distill neighborhood training via curriculum or importance sampling; cache reusable physics transforms to limit compute.
- *Operational tooling:* Package per-exam robustness profiles for model cards and site-readiness checklists; add *pre-deployment* synthetic probes matching local scanner settings.

## Potential Negative Societal Impact

C$^3$ could increase reliance on synthetic data; if misused without validity checks, this risks over-confidence and deployment to populations not represented in the original datasets. The framework should not replace external validation on real multi-site cohorts. Conservative validity gates and radiologist spot audits are recommended, and the use of synthetic counterfactuals should be disclosed in documentation and model cards to avoid accidental data leakage or privacy concer

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
