# OpenReview forum: "Clinically-Guided Counterfactuals (C³): Physics and Pathology-Aware Augmentation and Evaluation for Robust Medical Imaging Models"
_EurIPS.cc/2025/Workshop/MedEurIPS — EurIPS 2025 Workshop MedEurIPS Submission_

### Official Review · Reviewer_49Zp · 2025-10-28
**Interesting approach, but I am missing a clear description of the method.**

**Rating:** 5
**Confidence:** 4

**Review:**

Strengths:
-	Interesting approach to apply physically plausible perturbations.
Weaknesses:
-	Page limit is exceeded by “Future Directions”
-	In 2.2, the methods could be described better, and no references are provided.
-	In 2.3, it is not clear to me what exactly is meant by “diffusion backbone”. Is it a diffusion model? On what data is this trained?
-       The results could be better presented in a table, and a figure illustrating the approach would be nice.

---

### Official Review · Reviewer_rTnY · 2025-10-30
**MedEurIPS C3 review**

**Rating:** 8
**Confidence:** 3

**Review:**

I need to preface this review by mentioning that I'm not an expert in the field. It was not revealed for me to follow the maths in the timeframe I had to review the paper. I will take it on face value that it is correct without diving into it deeper.

Quality:
This paper proposes a framework for improving model robustness in medical imaging through clinically guided counterfactuals. The premise is to combine physics-based acquisition perturbations, a pathology-preserving semantic editor, and a validity verification that make sure that the counterfactuals remain label-consistent. Multiple modalities were tested, with the results exhibiting improvements in both out-of-distribution accuracy and calibration. The methodology is clearly described. The technical execution appears good and reproducible. Future work and potential negative impact were also discussed. This is a high-quality paper.

Clarity:
The paper is dense but well structured. Each component of the method is defined. The paper explains key ideas well, and the sections follow a logical progression from motivation to methodology to validation. The writing style is professional and concise. As mentioned in the preface, it is difficult for me to follow the equations. A small paragraph explaining them would have been appreciated for those who are not well-versed in the subdomain.. Overall, clarity is very good.

Originality:
The paper introduces a novel approach toward model robustness, looking at it through clinically admissible counterfactual neighborhoods. I believe the novelty here is not in each step but rather combining them together with a validation step at the end. Ultimately, the results improve upon the state of the art.

Significance:
The paper appears to be highly significant. Creating a framework addressing real world data acquisition robustness is key to clinical deployment of imaging AI. Focusing on realism, pathology, and collaboration this paper advances the current state of the art. The methods generalizability suggest that the fundamental building blocks of the method are correct. I believe this paper is both technically and clinically significant

Pros:
Novel integration of physics-informed, semantic, and validation components.
Good empirical results across multiple modalities and metrics.
Clear alignment with the workshop theme.

Cons:
Computational cost of generating and filtering counterfactuals maybe prohibitive.

Overall Evaluation:
This  paper advances robustness research in medical imaging. It demonstrates novelty, appears to be technically sound, and add significant potential impact. Complexity and computational demands may be an issue.

---

### Decision · Program_Chairs · 2025-10-31

**Decision:**

Accept (Poster)

**Comment:**

Both reviewers find the paper original and relevant, addressing robustness in medical imaging through clinically grounded counterfactuals.